# Pharmacological Activities and Characterization of Phenolic and Flavonoid Compounds in Methanolic Extract of *Euphorbia cuneata* Vahl Aerial Parts

**DOI:** 10.3390/molecules26237345

**Published:** 2021-12-03

**Authors:** Mohamed S. M. Soliman, Asmaa Abdella, Yehia A. Khidr, Gamal O. O. Hassan, Mahmoud A. Al-Saman, Rafaat M. Elsanhoty

**Affiliations:** 1Department of Industrial Biotechnology, Genetic Engineering and Biotechnology Research Institute, University of Sadat City (USC), Sadat City 32897, Egypt; mohamed.soliman@gebri.usc.edu.eg (M.S.M.S.); dr_aasmmaa@yahoo.com (A.A.); rafaat.elsanhoty@gebri.usc.edu.eg (R.M.E.); 2Department of Plant Biotechnology, Genetic Engineering and Biotechnology Research Institute, University of Sadat City (USC), Sadat City 32897, Egypt; yehia.khidr@gebri.usc.edu.eg; 3Medicinal and Aromatic Plants Department, Natural Products Unit, Desert Research Center, 1 Mathaf El-Matarya St., El-Matareya, Cairo 11753, Egypt; gamal_micro84@yahoo.com

**Keywords:** *Euphorbia cuneata* Vahl., antibacterial, antioxidant, cytotoxicity, cell lines, anti-inflammatory, phenolics, flavonoids

## Abstract

*Euphorbia cuneata* Vahl. (Euphorbiaceae) is a plant used in folk medicine for the treatment of pain and inflammation, although the biological basis for these effects has not been thoroughly investigated. The goal of this study was to investigate the pharmacological properties and characterization of phenolic and flavonoid compounds present in the aerial parts of *E. cuneata. E. cuneata* aerial parts were tested for antioxidant activity (DPPH), antibacterial activity, cell viability and cytotoxic effects, and anti-inflammatory activity. Phenolic and flavonoid contents (HPLC), and volatile constituents (GC-MS) were also characterized. The methanol extract had the highest antioxidant activity, while the ether extract had the lowest. The antioxidant activity of *E. cuneata* extract increased from (21.11%) at a concentration of 10 µg/mL to (95.53%) at a concentration of 1280 µg/mL. *S. aureus* was the most sensitive organism with the highest zone of inhibition and lowest MIC, with acetone extract; whereas *C. tropicalis* was the most resistant, with the lowest inhibition zone. MTT assay revealed that the methanol extract of *E. cuneata* had significant cytotoxic effects on the A549, Caco-2, and MDA-MB-231 cell lines, respectively. Lower concentrations of methanolic extract gave anti-inflammatory activity, and those effects were compared with indomethacin as a positive control. Pyrogallol was the most abundant phenolic acid, followed by caffeic, p-coumaric, ferulic, syringic, and gallic acids, respectively. The 7-hydroxyflavone and rutin flavonoids were also found in the extract. GC-mass analysis showed that aerial parts of *E. cuneata* were rich in methyl 12-hydroxy-9-octadecenoate. The volatile components were also composed of considerable amounts of hexadecanoic acid, methyl ester, (9E,12E)-octadeca-9,12-dienoyl chloride, and methyl octadeca-9,12-dienoate as well as a little amount of hexanal dimethyl acetal. It can be concluded that methanolic extract of *E. cuneata* could be used as an available source of natural bioactive constituents with consequent health benefits.

## 1. Introduction

Medicinal plants (e.g., *Vinca major*, *Zingiber officinale* and *Aloe vera*) are one of the most important sources of life-saving pharmaceuticals for the majority of the world’s population [1]. According to the World Health Organization (WHO), around three-quarters of the world’s population uses herbs and other traditional medicines to cure diseases, and there are several examples of new pharmaceuticals generated from wild plant species [2].

Gennari et al. [3] reported that cancer is one of the most life-threatening diseases in which deregulating proliferation of abnormal cells invades and disrupts surrounding tissues. Kelloff [4] also indicated that cancer is one of the most life-threatening diseases, with more than 100 different types occurring due to some molecular changes within the cell. It is the third leading cause of death worldwide following cardiovascular and infectious diseases.

Soliman et al. [5] mention that breast cancer is the most common form of cancer in women. Among the South Central Asian countries, the incidence of breast cancer is the highest in Pakistan [6]. It is the most frequent malignancy in women and accounts for 38.5% of all female cancers [5]. Moreover, the American Cancer Society stated that the second most common cause of cancer deaths in the US was colorectal cancer; furthermore, prostate cancer is the most common type of cancer diagnosed among men in the US [7].

Over time, it is increasingly being realized that many of today’s diseases are due to the “oxidative stress” that results from an imbalance between the formation and neutralization of prooxidants. Oxidative stress is initiated by free radicals, which seek stability through electron pairing with biological macromolecules such as proteins, lipids, and DNA in healthy human cells and cause protein and DNA damage along with lipid peroxidation [8]. These changes contribute to cancer, atherosclerosis, cardiovascular diseases, aging, and inflammatory diseases [9]. All cells are exposed to oxidative stress, and thus, oxidation and free radicals may be important in carcinogenesis at multiple tumor sites. The cyclooxygenase isoenzymes, COX-1 and COX-2 are the key enzymes involved in recruiting inflammation [10].

As a result, inflammation, free radicals, and carcinogenesis are closely related with one another. The drug candidates having anti-inflammatory and free radical scavenging activities are more appreciated as anticancer agents. Due to lack of effective drugs, cost of chemotherapeutic agents, and the side effects of anticancer drugs, cancer can be a cause of death. Therefore, efforts are still being made to search for effective naturally occurring anti-carcinogens that would prevent, slow, or reverse cancer development. Medicinal plants have a special place in the management of cancer [11]. It is estimated that plant-derived compounds in one or the other way constitute more than 50% of anticancer agents [12].

Many natural products discovered from medicinal plants, or secondary metabolites such as terpenoids, phenolic acids, lignans, tannins, flavonoids, quinones, coumarins, alkaloids, which exhibit significant antioxidant and other activities [13,14], have played an important role in treatment of cancer [15]. Studies by Kaur et al. [16] have shown that many of antioxidant compounds possess anti-inflammatory, antitumor, anti-mutagenic and anti-carcinogenic activities.

*Euphorbia cuneata* Vahl. one of the plant species in this genus, its native range is Africa and the Arabian Peninsula and has a variety of traditional uses in many countries [17]. Plants of the genus *Euphorbia* are widely distributed across temperate, tropical and subtropical regions of South America, Asia and Africa. *E. cuneata* has been shown to contain a variety of phytochemicals, including triterpenes (cyclocuneatol) and flavonoids (cuneatannin) [18]; and it does not contain the toxic diterpenes found in other Euphorbiaceae members [19]. Obesity, food poisoning, and constipation are all treated with stem juice combined with water or milk. Furthermore, the stem juice is applied to wounds and traumas in northern Yemen to stop bleeding. Analgesic and anti-inflammatory properties have also been reported [17].

The present study was designed to evaluate the anticancer, antimicrobial, anti-inflammatory, and antioxidant activities of extracts *E. cuneata* aerial parts, as well as to analysis of volatile constituents and phenolic acids by GC-Mass and HPLC.

## 2. Results and Discussion

### 2.1. DPPH Radical Scavenging Activity of E. cuneata Extracts

To find out more about *Euphorbia*’s possible health benefits, we tested the antioxidant capacities of different *E. cuneata* extracts. Table 1 shows that all *E. cuneata* extracts scavenged the DPPH radical in a dose-dependent manner. In terms of the effect of the extraction solvent on antioxidant activity, methanol extract had the highest scavenging potency (79.33%), followed by acetone (78.36%), and ethanol (75.76%), respectively. On the other side, among all extraction solvents, ether showed the lowest scavenging potency (42.31%).

It was noted that the scavenging potency increased with increasing *E. cuneata* concentration. The antioxidant activity was measured using the DPPH method with ascorbic acid as the standard. Ascorbic acid concentrations ranged from 1280 to 2.5 µg/mL. Ascorbic acid at a concentration of 2.5 μg/mL showed a % inhibition of 34.57% and for 1280 μg/mL 99.23%.

According to statistical analysis, the highest scavenging percentage (95.53%) was observed using the concentration of 1280 µg/mL that was followed by 640 µg/mL (91.16%), and 320 µg/mL (84.17%), respectively. On the other hand, the concentration of 10 µg/mL gave the lowest scavenging potency with an average of 21.11% value. The IC_50_ value of ascorbic acid was 10.6 μg/mL. The IC_50_ value for the methanolic extract was 21.5 μg/mL, 22.14 μg/mL for the acetone extract, and 25.8 μg/mL for the ethanol extract, respectively (Table 2).

The IC_50_ values for Euphorbia cuneata extracts were calculated using Graphpad Prism software (San Diego, CA, USA). IC50 was determined with a non-linear model.

Finally, there were no significant (*p* < 0.05) differences between the extracts of methanol, acetone, ethanol, water, ethyl acetate, and chloroform, according to statistical analysis.

Munro et al. [20] observed that the methanolic extract of *Euphorbia tirucalli* had better radical scavenging activity than the aqueous extract at all doses tested using the DPPH assay. In comparison to *E. peplus* and *E. cuneata* Shaker et al. [21] found that the antioxidant activities of *E. schimperiana* and their fractions (petroleum ether, ethyl acetate, and n-butanol/water) were the highest. Our findings, on the other hand, were in contrast to those of Awaad et al. [22] who found that the activity of the ethyl acetate extract was the highest and ranged from 26.51 to 83.50%; compared to 87.8% for the reference standard “ascorbic acid” at a concentration of 100 mM. However, we all agreed that the ether extract had the least % of activity among the extracts.

### 2.2. Antimicrobial Activity of E. cuneata Extracts

The antimicrobial activity of different *E. cuneata* extracts (water, methanol, acetone, ethanol, chloroform, ether, ethyl acetate, and methylene chloride) were tested against clinically isolated microorganisms that cause nosocomial respiratory tract infections (gram-negative bacteria: *Escherichia coli*, *Pseudomonas aeruginosa*, *Enterobacter cloacae*, and *Acinetobacter baumannii*; gram-positive bacteria: *Staphylococcus aureus*, *Streptococcus epidermidis*, and *Enterococcus faecalis*; and yeast: *Candida tropicalis*). Antimicrobial activity was determined qualitatively and quantitatively based on the existence of inhibitory zones, MIC, and MBC values.

The most active solvents on most isolates were acetone, chloroform, methanol, and ethyl acetate, whereas ether and ethanol had modest activity on three isolates. On the other hand, water and methylene chloride extracts had no inhibitory effect on isolated bacteria (Table 3). Based on earlier findings, several extracts of *E. cuneate* (acetone, chloroform, and ethyl acetate) showed the best activity on respiratory tract infections isolated bacteria, therefore we chose it for MIC and MBC testing. The best extract with a powerful MIC for *E. cuneate* was ethyl acetate, followed by acetone, and finally chloroform.

Gram-negative bacteria *E. cloacae* and *A. baumannii*; gram-positive bacteria: *S. aureus* and *E. faecalis*; and yeast: *C. tropicalis* had the lowest MIC of *E. cuneate* extracts acetone, chloroform, and ethyl acetate, as indicated in Table 3. *C. tropicalis* is the most resistance strains to all extracts. It showed the lowest zone of inhibition (13.50 mm) that was followed by *E. faecalis* which showed the same inhibition zone diameter. Furthermore, *S. aureus* was the most sensitive one where it scored the largest zone of inhibition (27.50 mm) and the least MIC (6.25 mg/mL).

Ethyl acetate, followed by acetone and chloroform, was the best extract with strong MIC and MBC. The MICs ranged from 6.25 to 25 mg/mL, while the MBCs of the ethyl acetate extract ranged from 6.25 to 100 mg/mL among the tested strains (Table 4).

Our findings support the findings of numerous studies, such as Zain et al. [23] who found that *E. cuneata* methanolic extract had potent antibacterial activity against Gram-positive and negative bacteria, and fungi. The MIC of fractions against the microorganisms under study varied from one species to the other. Kirbag et al. [24] also found that the antibacterial properties of methanolic extracts from particular *Euphorbia* species were the most effective against the tested microorganisms (*Staphylococcus aureus*, Cowan 1; *Bacillus megaterium*, DSM 32; *Proteus vulgaris*, FMC 1; *Klebsiella pneumonia*, FMC 5; *Escherichia coli*, ATCC 25922; *Pseudomonas aeruginosa*, DSM 50071; *Candida albicans*, FMC 17; *C. glabrata*, ATCC 66032; *C. tropicalis*, ATCC 13803; *Epidermophyton* sp., and *Trichophyton* sp.). *S. aureus* and *C. tropicalis* were the more sensitive microbes for all *Euphorbia* extracts. MIC of *Euphorbia* extracts toward the examined microorganisms varied from 6.25 mg/mL to 50 mg/mL, and this confirms our result. The results that obtained were also in agreement with the results that obtained by AL-Faifi [25] who found that *E. cactus* has antibacterial and anticancer properties. However, the findings were consistent with those of Li et al. [26] who found that *E. cuneata* contained many phytochemicals, including quercetin, kaempferol glycoside, and naringenin.

Gram-positive bacteria are more vulnerable to plant extracts than Gram-negative bacteria, according to Megeressa et al. [27], probably due to the nature of their cell walls. Gram-negative bacteria have phospholipid membranes with structural lipopolysaccharide components, making their cell walls antimicrobial-resistant.

### 2.3. Antiproliferative Effects on A549, Caco-2, and MDA-MB-231 Cells

Three different cancer cell lines (A549, Caco-2, and MDA-MB-231) were used to test the cytotoxicity of the methanolic *E. cuneata* extract only. The micro-culture tetrazolium (MTT) assay was used to measure the degree of cytotoxicity of the methanolic extract towards the cell lines. In comparison to the control, cytotoxic activity was expressed as a percentage of cell viability in A549, Caco-2, and MDA-MB-231 cell lines. The multiple concentrations of methanolic extract from *E. cuneata* were used and effective doses were calculated from dose-response curve. The dose-response curve was used to calculate effective doses using several concentrations of methanolic extract from *E. cuneata*.

The results of a cytotoxicity test of the *E. cuneata* extract against numerous cell lines are shown in Figure 1. The cytotoxicity of *E. cuneata* methanolic extract against the A549 cell line was investigated. Doxorubicin (reference standard) has an IC_50_ of 0.95 ± 0.21 µg/mL, while *E. cuneata* methanol extract has an IC_50_ of 118 ± 9.8 µg/mL (Figure 1A). The cytotoxicity of *E. cuneata* methanolic extract against the Caco-2 cell line was investigated. The IC_50_ value of Doxorubicin was 1.93 ± 0.24 µg/mL, while the IC_50_ value of *E. cuneata* methanol extract was 144 ± 17.2 µg/mL (Figure 1B). On the other side, cytotoxicity of methanolic extract of *E. cuneata* against MDA-MB-231 cell line was also evaluated. IC_50_ value of Doxorubicin was 0.64 ± 0.02 µg/mL and IC_50_ value of methanol extract of *E. cuneata* was 173.1 ± 16.3 µg/mL (Figure 1C).

Upon treatment with methanol extract of *E. cuneata*, the A549 cells showed an increased rate of cell death when compared to that in the Caco-2 and MDA-MB-231 cell lines at the same concentration of the extract used.

Previous studies have found an association between the antioxidant activity and anticancer properties of plant extracts [28], as well as evidence to suggest that the active constituents of *Euphorbia* species may have anticancer effects [29].

The *E. terracina* polar and polar fractions, as well as the *E. paralias* polar fraction were very active against human acute myeloid leukaemia (THP1) cells, according to Ben Jannet et al. [30]; with no cytotoxicity found against normal cells (CD14^+^ monocytes).

The same results were achieved by Al-Saraireh et al. [31] who analyzed the phytochemicals of different extracts of *E. hierosolymitana* and investigated their anticancer activity against a panel of different cancer cell lines. He found that butanol extract exhibited a significant selective concentration-dependent cytotoxicity in the colon cancer cell line (Caco-2) compared to normal cells (WI 38). This research studied the differential selectivity in anticancer activity between different extract of *E. cuneata* and holds significant promise for the development of new cancer therapy.

### 2.4. In Vitro Anti-Inflammatory Activity

The methanolic extract of *E. cuneata* showed considerable stabilization of RBCs membranes at various concentrations (1000, 500, 250, 125, 62.5, 31.25, 15.63, and 7.81 µg/mL). The percentage protection of methanolic extract at concentration 1000 µg/mL was higher than that of concentrations (71.25). However, at lower concentrations, the percentage protection was observed to be reduced. The results were tabulated in Figure 2. The IC_50_ value of indomethacin was 17.02 ± 0.91 μg/mL; however, the IC_50_ value for the methanolic extract was 73.8 ± 1.3 μg/mL.

According to several studies, *E. cuneata* is traditionally used to relieve pain and inflammation. *E. cuneata* has a substantial protective effect against lipopolysaccharide (LPS)-induced acute lung injury (ALI) in mice, according to Abdallah et al. [32]. This may be due to its antioxidant and anti-inflammatory properties. The presence of flavones, glucosides, and tannins in *E. hirta* extract contributes to its anti-inflammatory function by inhibiting NO, according to Shih et al. [33]; and various researchers have shown that increased NO production causes oxidative stress, DNA damage, and cell injury [34].

### 2.5. HPLC Identification of Phenolic Acids and Flavonoids 

A total of eleven different phenolic compounds consisting of five flavonoids (rutin, quercetin, kaempferol, luteolin, and 7-hydroxyflavone) and six phenolic acids (syringic acid, *p*-coumaric acid, caffeic acid, pyrogallol, gallic acid, ferulic acid) were investigated (Figure 3A,B).

Pyrogallol and caffeic acid were the most abundant phenolic compounds (50%) among all the phenolic acids and were detected in methanolic extract of *E. cuneata*. This was followed by *p*-coumaric acid, ferulic acid, syringic acid, and gallic acid, respectively (Table 5). 

7-hydroxyflavone (12.45%) and rutin (7.62%) were the most abundant flavonoids, followed by quercetin, luteolin, and kaempferol, respectively.

Using UHPLC-SRM/MS, Gopi et al. [35] found that ellagic acid, gallic acid, and quinic acid are the primary phenolics, whereas GC-MS analysis identified pyrogallol as the major constituent among the volatile components of the *E. hirta* methanolic extract. Flavonoids are plant-produced natural compounds with different phenolic structures. Munro et al. [20] observed that the water extract of *E. tirucalli* powder had 34.01 mg GAE (gallic acid equivalents)/g of total phenolic compounds, which was less than half of the methanol extract (77.33 mg GAE/g). Four flavonoidal substances isolated and identified from the alcohol extracts of *E. cuneate* Vahl. as naringenin, aromadendrin, apigenin, and 4’-O-methoxy-luteolin-7-O-rhamnoglucoside show antiulcerogenic potential, according to Awaad et al. [22]. Furthermore, naringenin, isoaromadendrin, dihydroquercetin, and isosinensin have all been isolated from *E. cuneata*, according to Bahar et al. [36]. According to our HPLC analyses, the methanolic extract of *E. cuneata* also included the flavonoid components rutin, kaempferol, and 7-hydroxyflavone.

### 2.6. GC-MS Analysis of E. cuneata Methanolic Extract

The volatile components of a methanolic extract of *E. cuneata* leaf were determined using GC-MS based on their retention time and peak area (Table 6). Five bioactive substances were identified and categorized based on their chemical structures using GC-MS analysis. Hexanal dimethyl acetal (6.59%), hexadecanoic acid, methyl ester (palmitic acid methyl ester) (21.34%), methyl octadeca-9,12-dienoate (13.47%), (9E,12E)-octadeca-9,12-dienoyl chloride (14.21%), and methyl 12-hydroxy-9-octadecenoate (44.39%) were the main volatile components.

The findings were consistent with those of İsmail Yener et al. [37], who found that linoleic acid, 17-tetratriacontane, palmitic acid, and hexatriacontane were the major fatty acids isolated from *E. aleppica*, *E. eriophora*, *E. macroclada*, *E. grisophylla*, *E. seguieriana* subsp. *seguieriana*, *E. craspedia*, *E. denticulata*, *E. falcata*, and *E. fistulosa*. Furthermore, according to Abdulselam Ertas et al. [38], palmitic acid was identified as the predominant component of fatty acid compositions obtained from the petroleum ether extracts of *E. gaillardii* and *E. macroclada*.

## 3. Materials and Methods

### 3.1. Chemicals

All chemicals and HPLC standards used were of high analytical grade, purchased from Sigma Chemical Co. (St. Louis, MO, USA).

### 3.2. Plant Materials and Sample Preparation

*Euphorbia cuneata* Vahl. (Family Euphorbiaceae) used in this study were collected in April and May 2019 from the Halayeb and Shalatayn (Jabal ‘Ilbah), Egypt. After washing, the aerial parts were air dried in the shade. The temperature ranged between 20–25 °C, with a relative humidity of 30–70% during the drying period (2 weeks). The content of water gradually decreased and became practically constant (about 8–9%), and then the leaf material was ground into a fine powder, and packed in screwed brown bottles until utilized in the following experiments.

#### Maceration Process

According to the method described by Romani et al. [39] 100 g of the powdered *E. cuneata* (aerial parts) was extracted by shaking at 150 rpm for 24 h at RT with 1 L of solvent (water, methanol, acetone, ethanol, chloroform, ether, ethyl acetate, and methylene chloride). In a Buchner funnel, extracts were filtered using Whatman No. 1 filter paper. The residue was re-extracted with 500 mL of solvent, filtered, and the filtrates were pooled and evaporated in a rotary evaporator at decreased pressure (Heidolph VV 2000). It was then dried in a desiccator under vacuum until it reached a consistent weight [40]. The weights of the final extracts were recorded, and the residue was re-suspended in the smallest amount of solvent possible to achieve a concentration of 10 mg/mL. The extracts were maintained at 4 °C until use. Triplicate determinations (*n* = 3) were applied in the study.

### 3.3. Antioxidant Capacity Assay

The 2,2-diphenyl-1-picrylhydrazyl (DPPH) assay [41] was used to determine the free radical scavenging activity. The violet color of DPPH was reduced to a pale-yellow tint in this assay due to the removal of hydrogen atoms from the antioxidant molecule. The DPPH reduction increase by increasing antioxidant activity was observed.

The assay reaction mixture consisted of 40 µL of *E. cuneata* extract at various concentrations (1280–10 µg/mL) generated by diluting the extract with the extraction solvent and 3 mL of methanolic solution of 0.1 mM (0.004%, *w*/*v*) DPPH radical. After vigorous shaking, the mixture was incubated at 37 °C for 30 min. A UV-visible spectrophotometer was used to detect the absorbance at 515 nm (Milton Roy, Spectronic 1201). The positive control was ascorbic acid. The following equation was used to calculate the relationship between decreasing absorbance of the reaction mixture and higher free radical scavenging activity: DPPH scavenging activity (%) = 100 × (A0 − A1)/(A0)(1)

The absorbances of the control and sample, respectively, are A0 and A1. The results are presented as the average of three replicate analyses, with the major values as well as the standard deviation (SD) provided.

The 50% inhibitory concentration (IC_50_), the concentration required to 50% DPPH radical scavenging activity was estimated from graphic plots of the dose response curve using Graphpad Prism software (San Diego, CA, USA).

### 3.4. Antimicrobial Activity

#### 3.4.1. Microorganism

Eight clinical microbial isolates were used in the experiments (Gram-negative bacteria: *Escherichia coli*, *Pseudomonas aeruginosa*, *Enterobacter cloacae*, and *Acinetobacter baumannii*; Gram-positive bacteria: *Staphylococcus aureus*, *Streptococcus epidermidis*, and *Enterococcus faecalis* and yeast: *Candida tropicalis*. The isolates were collected and identified from clinical specimens (sputum, end tracheal tube (ETT), nasal swab, and laryngeal swab (from respiratory tract infections patients)—from Giessen University Clinic—Germany [42].

#### 3.4.2. Paper disc Diffusion Assay

For disc diffusion assay, 100 µL of cell suspension was uniformly spread onto Mueller Hinton agar. The 6 mm diameter filter paper discs (Whatman No. 41) were positioned to contact the surface of infected agar and were impregnated with 25 µL extract made at a concentration of 10 mg/mL. The plates were incubated for 24 h at 37 °C. The diameter of the zone of inhibition (ZOI) was then measured precisely, and the means of the triplicates were determined [43].

#### 3.4.3. Determination of Minimal Inhibitory Concentration (MIC)

Each of the *E. cuneata* extracts (acetone, chloroform, and ethyl acetate) was tested against the studied bacterial isolates as a quantitative experiment to determine minimal inhibitory concentrations (MIC). The bifold dilution method [44] was employed using a series of 6 tubes holding 100 µL of Mueller Hinton broth (Accumix–Verna, India) to achieve final dilutions of 100, 50, 25, 12.5 6.25, and 3.12% (*v*/*v*). The isolated microorganisms were inoculated with standard bacterial inoculums (5 × 10^5^).

Microbial isolates were inoculated into all 6 dilutions post through different fractions. The inoculation tubes were incubated at 37 °C overnight. The MIC value of these fractions against the tested microorganisms’ isolates was calculated using the greatest dilution of the tested different extracts of *E. cuneata* to inhibit growth (no turbidity in the tube) [45].

#### 3.4.4. Determination of Minimal Bactericidal Concentration (MBC)

One hundred µL from tubes that showed no signs of growth or turbidity was inoculated into sterilized Mueller Hinton agar plates by streak plate method, which was then incubated at 37 °C for an overnight period. The MBC value of the different tested extracts (acetone, chloroform, and ethyl acetate) was obtained by the lowest concentration that did not demonstrate any growth of the tested microorganisms [46].

### 3.5. Cell Viability and Cytotoxic Effects

#### 3.5.1. Mammalian Cell Lines

The human lung cancer cell line (A549), the human colon adenocarcinoma cell line (Caco-2), and the human breast cancer cell line (MDA-MB-231) were obtained from the American Type Culture Collection (ATCC, Rockville, MD, USA) by personal communication with Prof. Dr. Mahmoud Al-Aeser, the Regional Center for Mycology and Biotechnology (RCMB) at Al Azhar University, Egypt.

#### 3.5.2. Propagation of Cell Lines

The cells were propagated in Dulbecco’s modified Eagle’s medium (DMEM) supplemented with 10% heat-inactivated fetal bovine serum (FBS), 1% l-glutamine, HEPES buffer and 50 µg/mL gentamycin. All cells were maintained at 37 °C in a humidified atmosphere with 5% CO_2_ and were sub-cultured two times a week.

The cells were grown in Dulbecco’s modified Eagle’s medium (DMEM), which contained 10% heat inactivated foetal bovine serum (FBS), 1% l-glutamine, HEPES buffer, and 50 µg/mL gentamycin. All cells were kept at 37 °C in a humidified environment with 5% CO_2_ and were sub-cultured twice a week.

#### 3.5.3. Cytotoxicity Assay

Cytotoxicity assay was evaluated according to Mosmann [47] the cells were seeded in 96-well plate at a cell concentration of 1 × 10^4^ cells per well in 100 µL of growth medium. Fresh medium containing different concentrations of the methanolic extract was added after 24 h of seeding. Serial two-fold dilutions of the methanolic extract were added to confluent cell monolayers dispensed into 96-well, flat-bottomed microtiter plates (Falcon, NJ, USA) using a multichannel pipette. The microtiter plates were incubated at 37 °C in a humidified incubator with 5% CO_2_ for a period of 24 h. Three wells were used for each concentration of the test sample. Control cells were incubated without extract and with or without DMSO. The little % of DMSO present in the wells (maximal 0.1%) was found not to affect the experiment.

After incubation of the cells at 37 °C for 24 h, the viable cells yield was determined by a colorimetric method. In brief, after the end of the incubation period, media were aspirated and the crystal violet solution (1%) was added to each well for at least 30 min. The stain was removed and the plates were rinsed using tap water until all excess stain was removed. Glacial acetic acid (30%) was then added to all wells and mixed thoroughly, and then the absorbance of the plates were measured after gently shaken on microplate reader (TECAN, Inc., Morrisville, NC, USA), using a test wavelength of 490 nm. All results were corrected for background absorbance detected in wells without added stain. Treated samples were compared with the cell control in the absence of the tested compounds. All experiments were carried out in triplicate. The cell cytotoxic effects of different concentration of methanolic extract were calculated. The optical density was measured with the microplate reader (SunRise, TECAN, Inc, Morrisville, NC, USA) to determine the number of viable cells and the % of viability was calculated as [(OD_T_/OD_C_)] × 100%.

Where OD_T_ is the mean optical density of wells treated with the tested sample and OD_C_ is the mean optical density of untreated cells. The relation between surviving cells and drug concentration is plotted to get the survival curve of each tumor cell line after treatment with the specified compound. The 50% inhibitory concentration (IC_50_), the concentration required to cause toxic effects in 50% of intact cells, was estimated from graphic plots of the dose response curve for each concentrate using Graphpad Prism software (San Diego, CA, USA).

### 3.6. In Vitro Anti-inflammatory Activity

#### 3.6.1. Erythrocyte Suspension Preparation

The anti-inflammatory efficacy has been studied in vitro using the membrane stabilization method. Red blood cells (RBCs) were collected from a healthy human volunteer who had not taken any non-steroidal anti-inflammatory drugs for two weeks prior to the study. The blood was washed three times with isotonic buffered solution (154 mM NaCl) in 10 mM sodium phosphate buffer (pH 7.4). The blood was centrifuged for 10 min at 3000× *g*.

#### 3.6.2. Hypotonic Solution-Induced Erythrocyte Haemolysis

Membrane stabilizing activity of the methanolic extract was assessed using hypotonic solution-induced erythrocyte haemolysis. The sample consisted of stock erythrocyte (RBCs) suspension (0.50 mL) mixed with 5 mL of hypotonic solution (50 mM NaCl) in 10 mM sodium phosphate buffered saline (pH 7.4) containing the methanolic extract (1000–7.81 µg/mL) or indomethacin (as positive control). The negative control sample consisted of 0.5 mL of RBCs mixed with hypotonic-buffered saline solution alone. The mixtures were incubated for 10 min at room temperature and centrifuged for 10 min at 3000× *g*. In 96 well plates, the absorbance of the supernatant was measured at 540 nm. The percentage inhibition of haemolysis or membrane stabilization was calculated according to modified method described by Shinde et al. [48].
Inhibition of heamolysis (membrane stabilization%) = OD_1_ − OD_2_/OD_1_ × 100(2)
where OD_1_ = Optical density of hypotonic-buffered saline solution alone; OD_2_ = Optical density of extract in hypotonic solution.

The IC_50_ value was defined as the concentration of the sample to inhibit 50% RBCs haemolysis under the assay conditions.

### 3.7. HPLC Identification of Phenolic and Flavonoid Compounds of E. cuneata Methanolic Extract

HPLC (Agilent 1100, Palo Alto, CA, USA) was employed for the detection of phenolic acids and flavonoids; consists of two LC pumps, a UV/V detector, and a C18 column (150 mm × 4.60 mm, 5 µm particle size). The Agilent Chem Station was used to obtain and analyze chromatograms. Phenolic acids were separated using a gradient mobile phase consisting of two solvents: solvent A (methanol) and solvent B (acetic acid in ultrapure water, 1:25). Elution from the column was achieved with the following gradients: 0 to 3 min of solvent B. This was followed by 50% eluent A for the next 5 min, after which the concentration of A was increased to 80% for the next 2 min and then reduced to 50% again for the following 5 min.

Flavonoids were separated using a gradient mobile phase consisting of two solvents: solvent A (acetonitrile) and solvent B (0.2%, *v*/*v* aqueous formic acid) with an isocratic elution (70: 30) program [49,50].

Before injection, the extract was filtered (0.45 μm) and diluted with methanol. The applied flow rate was mL/min. The compounds were identified by comparing their retention periods and absorption spectra with those obtained for the analyzed standards, and were quantified using external calibration curves of reference standards from their peak areas at 280 and 320 nm for phenolic compounds and flavonoids, respectively.

### 3.8. Determination of the Volatile Components of E. cuneata Methanolic Extract

The compounds were identified by WILEY & NIST MASS SPECTRAL DATA BASE. Authentic chemicals were also used in identification when they were available.

A Thermo Scientific TRACE 1310 Gas Chromatograph was used in conjunction with an ISQ LT (single quadrupole Mass Spectrometer) for the GC-MS analysis of the methanolic extract. DB5-MS, 30 m; 0.25 mm ID (J&W Scientific, Folsom, CA, USA) was the column. The carrier gas was helium at a flow rate of 1.0 mL/min. Starting at 40 °C and maintaining that temperature for 3 min; increasing to 280 °C with a 5.0 °C/min heating rate and maintaining that temperature for 5 min; and increasing to 290 °C with a 7.5 °C/min heating rate and maintaining that temperature for 1 min. Temperatures of 200 °C for injection and 300 °C for detection were used. Electron ionization (EI) was used to obtain mass spectra at 70 eV, using a spectral range of *m*/*z* 40–450. The compounds were identified by WILEY & NIST MASS SPECTRAL DATA BASE. Authentic chemicals were also used in identification when they were available. 

The identities of some components were confirmed by mass spectral and retention data of the authentic chemicals obtained under identical GC-MS conditions, and the concentration of the selected compounds was calculated based on the standard calibration curve.

### 3.9. Statistical Analysis

An analysis of variance was performed on all of the data (ANOVA). The main values, and the standard deviation (SD), were calculated from three samples of each item. Duncan’s multiple range tests (*p* ≤ 0.05) were used to examine the significance of the variable mean differences. SPSS 16 was used to conduct all of the analyses [51].

## 4. Conclusions

Several analyses in this study revealed that the aerial parts of *E. cuneata* Vahl are rich in bioactive compounds. The methanol extract of *E. cuneata* had the highest scavenging potency (79.33%) at a concentration of 10 µg/mL, and the best extract with a powerful MIC was ethyl acetate, followed by acetone, and lastly chloroform. The methanolic extract showed sufficient membrane stability to be considered an anti-inflammatory agent. In vitro models revealed that the methanol extract exhibited potent anticancer activity in several cancer cell lines. The main phenolic acids in the methanolic extract were pyrogallol, followed by caffeic, *p*-coumaric, ferulic, syringic, and gallic acids; and the main flavonoids were 7-hydroxyflavone, rutin, and quercetin, respectively. GC–MS analysis revealed that methyl 12-hydroxy-9-octadecenoate was the most abundant component, with a value of approximately 44.39%. Collectively, this study revealed the potent protective effect of the methanolic extract of *E. cuneata* against pathogenic microorganisms and several human cancer cell lines, which may be linked to its antioxidant and anti-inflammatory activities.

It can be concluded that methanolic extract of *E. cuneata* could be employed as a readily available source of natural bioactive compounds with medicinal benefits, which would be particularly beneficial to the pharmaceutical industry. Futures perspectives of operating an in vivo study and exploration of *E. cuneata* as a natural pharmacological agent to replace chemical drugs could be a promising step.

## Figures and Tables

**Figure 1 molecules-26-07345-f001:**
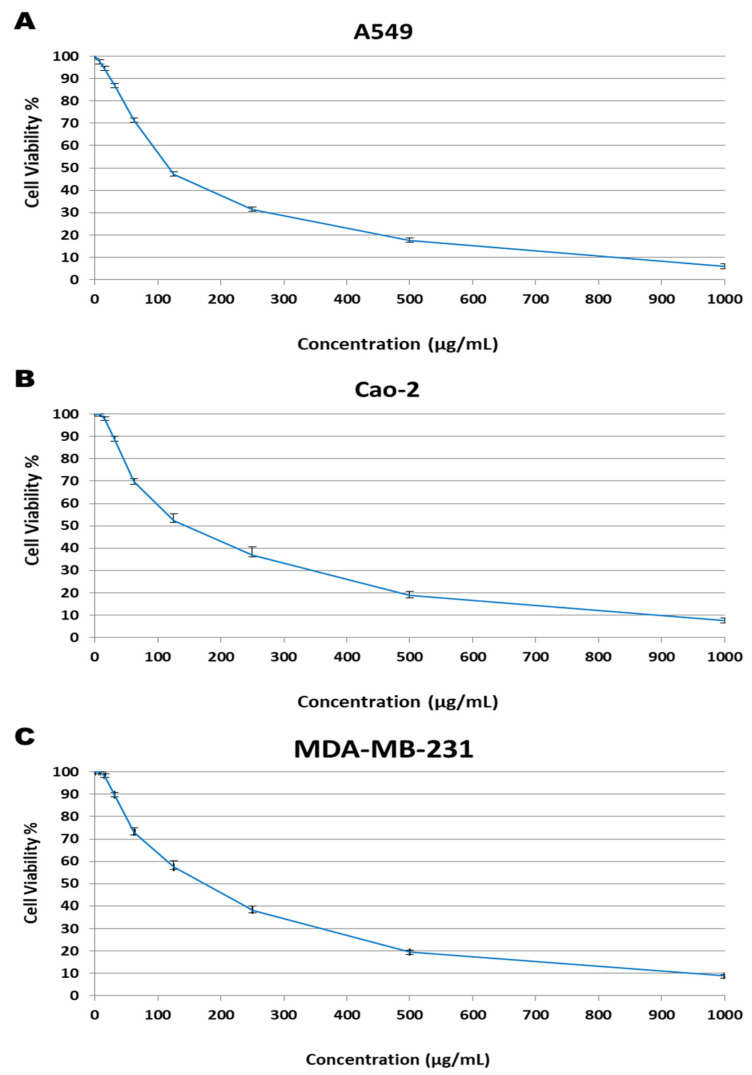
Toxicity effects of the *Euphorbia cuneata* leaf methanolic extract against cancer cell line: (**A**) A549, (**B**) Caco-2, (**C**) MDA-MB-231 after 24 h of incubation.

**Figure 2 molecules-26-07345-f002:**
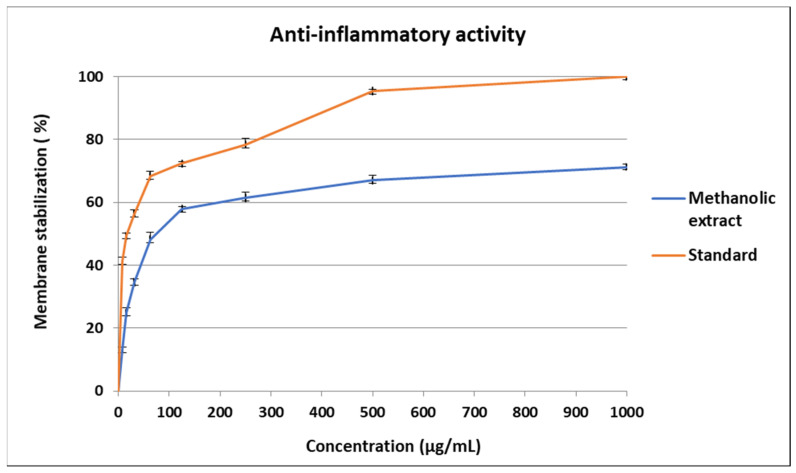
In vitro anti-inflammatory activity of methanolic extract of *Euphorbia cuneata* leaf.

**Figure 3 molecules-26-07345-f003:**
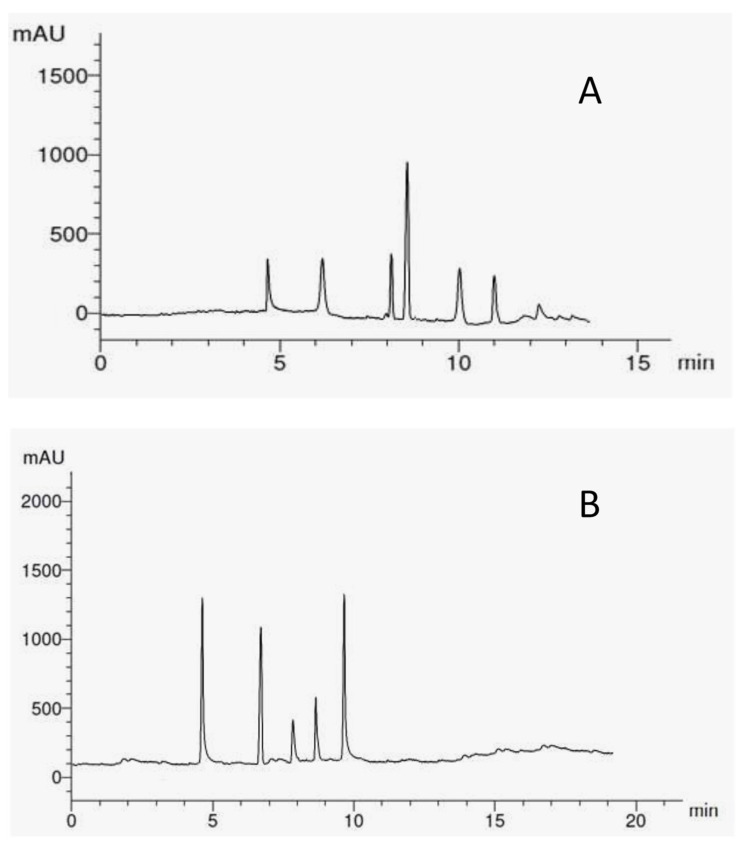
HPLC chromatographic profiles of phenolic acids (**A**) and flavonoids (**B**) in methanolic extract of *Euphorbia cuneata* leaf monitored at 280 and 320 nm, respectively.

**Table 1 molecules-26-07345-t001:** Radical scavenging activity of *Euphorbia cuneata* extracts at different concentrations toward DPPH.

Radical Scavenging Activities of Extracts *									
Conc. (µg/mL)	Water	Acetone	Chloroform	Methylene Chloride	Ether	Methanol	Ethanol	Ethyl Acetate	Conc. Mean ± SE
1280	98.56 ± 0.92	98.78 ± 0.64	93.76 ± 0.62	92.06 ± 1.42	87.96 ± 1.42	98.17 ± 0.81	97.91 ± 1.35	97.06 ± 0.82	95.53 ± 03.66 ^a #^
640	97.08 ± 0.74	97.15 ± 0.59	90.82 ± 0.74	78.47 ± 1.85	78.35 ± 1.93	96.76 ± 0.52	95.82 ± 0.94	94.82 ± 1.34	91.16 ± 07.60 ^ab^
320	94.63 ± 1.35	95.06 ± 0.92	77.06 ± 1.93	69.18 ± 1.94	60.18 ± 1.26	95.18 ± 0.74	94.06 ± 0.72	88.04 ± 0.78	84.17 ± 12.81 ^bcd^
160	89.47 ± 1.09	91.29 ± 0.83	65.88 ± 2.14	58.35 ± 3.17	46.94 ± 3.98	93.88 ± 0.68	91.53 ± 0.65	79.12 ± 0.56	77.06 ± 16.71 ^cd^
80	78.59 ± 2.47	87.53 ± 1.75	52.47 ± 2.95	47.76 ± 2.82	29.18 ± 1.74	90.39 ± 0.53	86.71 ± 1.83	73.06 ± 1.32	68.21 ± 21.00 ^de^
40	59.88 ± 2.94	81.04 ± 1.62	39.84 ± 3.62	37.65 ± 1.91	19.29 ± 0.63	80.62 ± 1.84	74.59 ± 2.73	56.35 ± 1.97	56.16 ± 21.03 ^e^
20	39.76 ± 2.32	46.35 ± 2.19	28.12 ± 1.76	32.47 ± 1.35	13.41 ± 0.57	47.56 ± 2.98	39.88 ± 2.94	35.94 ± 1.82	35.44 ± 10.32 ^f^
10	20.34 ± 1.75	29.71 ± 1.35	20.94 ± 0.82	18.92 ± 1.46	03.18 ± 0.26	32.12 ± 1.74	25.63 ± 1.75	18.06 ± 2.95	21.11 ± 08.30 ^g^
*Group mean ± SE*	72.29 ± 27.50 ^ab^	78.36 ± 24.23 ^a^	58.61 ± 25.00 ^abc^	54.36 ± 23.17 ^bc^	42.31 ± 29.14 ^c^	79.33 ± 23.67 ^a^	75.76 ± 25.98 ^ab^	67.81 ± 26.86 ^abc^	

* Radical scavenging activity given as percentage inhibition. The percentage inhibition value of the standard compound ascorbic acid was 99.23% at a concentration of 1280 µg/mL. # ^a–g^ Means followed by the same letter(s) within a column or row are not significantly different (*p* ≤ 0.05) according to Duncan’s multiple range test.

**Table 2 molecules-26-07345-t002:** IC_50_ values of *Euphorbia cuneata* extracts toward DPPH.

Type of Extract	IC_50_ (µg/mL)
Water	30.22 ± 2.86
Acetone	22.14 ± 1.98
Chloroform	72.26 ± 5.42
Methylene cholide	96.91 ± 6.27
Ether	197.1 ± 14.3
Methanol	21.50 ± 1.80
Ethanol	25.80 ± 2.10
Ethyl acetate	33.80 ± 2.83

**Table 3 molecules-26-07345-t003:** Antimicrobial activity of different *Euphorbia cuneata* extracts against respiratory tract infection isolates measured as zone of inhibition diameter (ZOI, mm).

Isolated Microorganisms	Extract								
	Water	Methanol	Acetone	Ethanol	Chloroform	Ether	Ethyl Acetate	Methylene Chloride	Gentamicin (10 µg/mL)
*S. aureus*	NI **	24.5 ± 0.7 *	27.5 ± 3.5	17.0 ± 1.4	19.0 ± 1.4	25.0 ± 0.0	21.5 ± 4.9	NI	26.0 ± 0.0
*S. epidermidis*	NI	20.0 ± 0.0	16.0 ± 1.4	NI	20.0 ± 7.0	25.0 ± 0.0	20.0 ± 1.4	NI	25.0 ± 0.0
*E. faecalis*	NI	22.5 ± 3.5	19.0 ± 1.4	13.5 ± 0.7	NI	NI	NI	NI	25.0 ± 0.0
*E. coli*	NI	16.5 ± 0.7	19.5 ± 3.5	NI	NI	25.0 ± 0.0	NI	NI	-
*E. cloacae*	NI	18.5 ± 2.1	22.5 ± 0.7	NI	16.5 ± 2.1	NI	17.5 ± 3.5	NI	-
*P. aeruginosa*	NI	21.0 ± 1.4	16.0 ± 2.8	NI	18.5 ± 0.7	NI	21.5 ± 4.9	NI	-
*A. baumannii*	NI	25.0 ± 7.0	15.5 ± 2.1	25.0 ± 0.0	18.5 ± 4.9	NI	18.0 ± 4.2	NI	-
*C. tropicalis*	NI	13.5 ± 0.7	16.5 ± 0.7	NI	22.5 ± 0.7	NI	16.0 ± 2.8	NI	-

* Zone of inhibition (ZOI) diameters included the diameter of paper disc (6 mm) ± standard deviations. ** NI: no inhibitory effect.

**Table 4 molecules-26-07345-t004:** The values of MIC and MBC for the acetone, chloroform, and ethyl acetate extracts of *Euphorbia cuneata* against respiratory tract infection isolates.

Isolated Microorganisms	Extract					
	MIC * (mg/mL)			MBC ** (mg/mL)		
	Acetone	Chloroform	Ethyl Acetate	Acetone	Chloroform	Ethyl Acetate
*S. aureus*	6.25	6.25	6.25	12.5	12.5	12.5
*S. epidermidis*	50.0	50.0	25.0	100	100	100
*E. faecalis*	6.25	NI	NI	12.5	NI	NI
*E. coli*	25.0	50.0	25.0	100	100	100
*E. cloacae*	6.25	6.25	6.25	12.5	12.5	12.5
*P. aeruginosa*	25.0	50.0	25.0	100	100	100
*A. baumannii*	6.25	6.25	6.25	12.5	12.5	12.5
*C. tropicalis*	6.25	6.25	6.25	12.5	6.25	6.25

* MIC: Minimum inhibitory concentration, ** MBC: Minimum bactericidal concentration.

**Table 5 molecules-26-07345-t005:** Phenolic acids and flavonoids composition of methanolic extract of *Euphorbia cuneata*.

NO.	Retention Time (min)	Compound Name	Peak Area%
1	04.89	syringic acid	05.12
2	06.00	*p*-coumaric acid	06.33
3	08.00	caffeic acid	07.41
4	08.80	pyrogallol	13.63
5	10.00	gallic acid	04.98
6	11.20	ferulic acid	05.17
7	04.80	rutin	07.62
8	06.90	quercetin	06.99
9	08.00	kaempferol	03.25
10	08.80	luteolin	04.14
11	10.00	7-hydroxyflavone	12.45

**Table 6 molecules-26-07345-t006:** Volatiles composition of methanolic extract of *Euphorbia cuneata* leaf.

Compound Name	Retention Time (min)	Molecular Formula	*m*/*z* Fragments	Peak Area Percentage *
Hexanal dimethyl acetal [1,1-Dimethoxyhexane]	5.93	(C_8_H_18_O_2_) 146.23	45, 55, 75 ^#^, 115	6.59
Hexadecanoic acid, methyl ester [palmitic acid methyl ester]	29.13	(C_17_H_34_O_2_) 270	74 ^#^, 87, 129, 143	21.34
Methyl octadeca-9,12-dienoate	32.35	(C_19_H_34_O_2_) 294.5	55, 67 ^#^, 81, 95	13.47
Linoleoyl chloride [(9E,12E)-octadeca-9,12-dienoyl chloride]	32.46	(C_18_H_31_ClO) 298.9	41, 55 ^#^, 74, 83, 87	14.21
Methyl 12-hydroxy-9-octadecenoate	35.97	(C_19_H_36_O_3_) 312.5	55 ^#^, 69, 74, 97, 166	44.39

* The most intense ion is indicated by the base peak (tallest peak). ^#^ (Peak area of individual volatile compound/total peak areas of all volatiles) × 100.

## Data Availability

The data presented in this study are available in this manuscript.

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
