# Peer review of "Pharmacological Activities and Characterization of Phenolic and Flavonoid Compounds in Methanolic Extract of Euphorbia cuneata Vahl Aerial Parts"

_molecules, 2021, doi:10.3390/molecules26237345_

Round 1
Reviewer 1 Report
Manuscript titled: Pharmacological Activities and Characterization of Phenolic and Flavonoid Compounds in Methanolic Extract of Euphorbia cuneata Vahl Aerial Parts is an interesting study on the bioactive and antibiotic properties of Euphorbia cuneata extracts. The research described in the paper was well planned, the paper contains an interesting methodology, which is reflected in the achievement of many good results. The work has been properly prepared, however, before possible publication, please correct the issues listed below.
Abstract: abstract should contain an introduction - background, which will introduce the reader to the research problem presented in the manuscript.
Introduction:
Medical plants: please give some examples of commonly known medicinal plants.
"...there are several examples of new pharmaceuticals generated from plant sources [2], some of which come from wild plant species whose
properties were not yet known [3]..." - are pharmaceuticals actually produced from plants of unknown properties? Please explain this or correct this sentence.
Is the plant Euphorbia cuneata approved for human consumption? Does it not contain toxic compounds that may be harmful to health?
There is no citation in the last paragraph of the introduction. Please insert literature sources confirming the data described before the last sentence in the introduction.
Materials and Methods
Plant Materials and Sample Preparation:
Please specify the conditions for drying the plant, i.e. time and temperature of the process. Please provide the activity of the water for which the plant was dried, the water content after the drying process is completed. These are very important parameters determining the mycobiological quality of the obtained dried material and thus its durability. Why was the plant stored at -20 C after the drying process?
Please remove figure 1 from manuscript. There is no need to provide a plant distribution map.
Please add a title for the paragraph described under Figure 1. If I understand correctly, this is a description of the extraction of active compounds from the tested plant? Please describe the extraction method in more detail, especially the final section on the dissolution of the extracted compounds - in which solvents were these compounds dissolved.
Results and Discussion: please try to analyze the relationship between the marked features of the tested plant. Is there a statistically significant co-proportional relationship between the marked features of the plant?
GC-MS Analysis of E. cuneata Methanolic Extract: there is no discussion in this part of the manuscript. Based on the analysis of volatile compounds and literature data, is it possible to state the nature of the compounds present in the plant under study? Positive or negative perception by consumers?
Conclusions: Please complete the conclusion or, if necessary, add in the discussion the practical application of the tested plant. Based on the results obtained in the study, what dose could be administered to obtain a therapeutic effect after consuming the infusion of the tested plant? Should human or animal studies be performed to confirm the medicinal properties of the test plant?
Author Response
"Please see the attachment."

Reviewer 2 Report
The manuscript represents pharmacological activities and characterization of phenolic and flavonoid compounds in methanolic extract of euphorbia cuneata vahl aerial part. Authors have done detailed study of methanolic extract and presented the data concisely. There few points needs to address. - It would be nice if authors could talk about previous studies done on extracts of any parts of E. cuneata for anticancer, antimicrobial, anti-inflammatory, and antioxidant activities. - Authors should talk about the reason behind using ascorbic acid as a control in antioxidant capacity assay. Have authors achieved the same activity of ascorbic acid which was reported previously? Similarly Indomethacin was selected a standard for anti-inflammatory activities, required to mention the reason behind it. - Authors have identified phenolic acids and flavanoids compounds in the extract but it would be nice if authors could test or hypothesize which compounds are the potential pharmacologically active for anti-oxidative or antiinflammatory activities which very important for the final goal of the current manuscript.Author Response
"Please see the attachment."

This manuscript is a resubmission of an earlier submission. The following is a list of the peer review reports and author responses from that submission.
Round 1
Reviewer 1 Report
The authors propose a manuscript titled “Pharmacological Activities and Characterization of Phenolic and Flavonoid Compounds in Methanolic Extract of Euphorbia cuneata Vahl Aerial Parts”
The article is original, well structured and written. In particular, this study takes into consideration the pharmacological properties and characterization of phenolic and flavonoid compounds present in the aerial parts of Euphorbia cuneata Vahl. In particular were tested the aerial parts for antioxidant activity (DPPH), antibacterial activity, cell viability and cytotoxic effects, and anti-inflammatory activity. Phenolic and flavonoid contents (HPLC), and volatile chemicals (GC-MS) were also characterized. In the specific with MTT assay revealed that the methanol extract of E. cuneata had significant cytotoxic effects on the A549, Caco-2, and MDA-MB-231 cell lines, respectively, while lower concentrations of methanolic extract gave anti-inflammatory activity, and those effects were compared with indomethacin as a positive control. The authors conclud that methanolic extract of E. cuneata could be used as an available source of natural bioactive constituents with consequent health benefits.
I read the work carefully in a critical way, and suggested in detail some concepts only in order to further improve the good work done. In particular, some concepts need to be referenced, indicating point by point where is necessary.
Abstract. Please summarize, not report a specific data with percentage ….
- Introduction
Well done, the concept are correct but in some case need to complete in the correct way. The suggestions are in bold:
- Lines 40-43. According to the World Health Organization (WHO), around three-quarters of the world's population uses herbs and other traditional medicines to cure diseases, and there are several examples of new pharmaceuticals generated from plant sources [2], some of which come from wild plant species whose properties were not yet known [e.g. Perrino et al. 2021…];
A references to be added:
Perrino, E.V.; Valerio, F.; Jallali, S.; Trani, A.; Mezzapesa, G.N. Ecological and Biological Properties of Satureja cuneifolia and Thymus spinulosus Ten.: Two Wild Officinal Species of Conservation Concern in Apulia (Italy). A Preliminary Survey. Plants 2021, 10, 1952. https://doi.org/10.3390/plants10091952.
- Lines 51-53. “The incidence of breast cancer is the highest in Pakistan among the South Central Asian countries [choose a reference]”. It is the most frequent malignancy in women and accounts for 38.5% of all female cancers [choose a reference].
- Lines 58-61. “Oxidative stress is initiated by free radicals, which seek stability through electron pairing with biological macromolecules such as proteins, lipids, and DNA in healthy human cells and cause protein and DNA damage along with lipid peroxidation [choose a reference].
- Lines 71-72. “Medicinal plants have a special place in the management of cancer [choose a reference]”.
- Lines 74-77. “Many natural products discovered from medicinal plants, or secondary metabolites such as terpenoids, phenolic acids, lignans, tannins, flavonoids, quinones, coumarins, alkaloids, which exhibit significant antioxidant and other activities [g. Valerio et al. 2021, choose other references], have played an important role in treatment of cancer [choose a reference].
- Lines 80-85. Why in italics some periods, there is no reason! Correct in the suggest way: “Euphorbia cuneata Vahl one of the plant species in this genus, its native range is Africa and the Arabian Peninsula and has a variety of traditional uses in many countries [11]. Plants of the genus Euphorbia are widely distributed across temperate, tropical and subtropical regions of South America, Asia and Africa. E. cuneata has been shown to contain a variety of phytochemicals, including triterpenes (cyclocuneatol) and flavonoids (cuneatannin) [12]; and it does not contain the toxic diterpenes found in other Euphorbiaceae members.
A references to be added:
Valerio, F.; Mezzapesa, G.N.; Ghannouchi, A.; Mondelli, D.; Logrieco, A.F.; Perrino, E.V. Characterization and Antimicrobial Properties of Essential Oils from Four Wild Taxa of Lamiaceae Family Growing in Apulia. Agronomy 2021, 11, 1431. https://doi.org/10.3390/agronomy11071431.
- Materials and methods
- Lines 94-95. Please give a georeferenced map of Halaib and Shalatin (Gabal Elba) in Egypt.
- Lines 103-106. Please give a reference for method used or is new for the proposed study? “It was then dried in a desiccator under vacuum until reached a consistent weight. The weights of the final extracts were recorded, and the residue was re-suspended in the smallest amount of solvent possible to achieve a concentration of 10 mg/mL. The extracts were maintained at -4ËšC until use. Triplicate determinations (n=3) were applied in the study [give a method or better clarify]”.
- Lines 249-253. The authors declare that for the statistical analysis and in particular for the variance was adopted ANOVA, but only 3 samples of each item! A higher number of repetitions would have been better, even if 3 samples is the minimum allowed for this type of analysis.
- Results and Discussion
Well done, the figures and tables are clear. No observations.
- particular area is fluid as field exploration and taxonomic studies progress…”
Please follow the guidelines of the journal about references
Author Response
"please see the attachment file"

Reviewer 2 Report
The Authors propose a phytochemical analysis and biological evaluation (antioxidant, antibacterial and anti-proliferative) of extracts from aerial parts of Euphorbia cuneata.
The results are in my opinion too preliminary and sometime inconsistent to deserve publication in Antioxidants.
Only one in vitro assay was used to characterize antioxidant activity. Several in vitro assays are generally required and cellular assay are also needed for such a Journal.
The selection of methanolic extract is unclear since there are no significant statistical differences between methanol and other solvents such as ethanol, acetone and maybe water. Statistical analysis has to been provided such each assay.
Antiproliferative assay is very basic and not sufficient to determine any potential anti-cancer action.
MBC have to be determined as well and the type of antimicrobial action determined as well.
Results presented in Figure 3 are unclear for me. Detection was done at 280 nm for both types of compounds. How can you differentiate phenolic compounds to flavonoids in your extract under those conditions? The gradients are different but the compounds are all present in your extract.
Other minor corrections:
Bacteria and plant scientific names have to be italicized (e.g., line 21: S. aureus)
Line 92: (St. Louis, MO, USA)
Line 151: different front
Line 164: L-glutamine, use small capital letter for L
Line 200: origin of blood cells. Any ethical declaration needed?
Line 350: Antiproliferative Effects on
Author Response
"please see the attachment file"

Reviewer 3 Report
The aim of this study was to investigate the pharmacological properties and characterization of phenolics present in the aerial parts of Euphorbia cuneata Vahl. E. cuneata aerial parts were tested for antioxidant activity (DPPH), antibacterial activity, cell viability and cytotoxic effects, and anti-inflammatory activity.
The authors state that:
L: 17-Phenolic and flavonoid contents (HPLC), and 17 volatile chemicals (GC-MS) were also characterized.
Please specify in "Materials and Methods" how the quantitative determination of phenolic compounds was done. Reporting only peak areas is not a relevant method. Usually the quantitative determinations are made based on calibration curves. Please clarify this / specify in "Materials and Methods".
Also, the method of determining the concentration of volatile compounds must be specified in "Materials and Methods"
Please be very careful! The authors state: L 85-88:
''The present study was designed to evaluate the anticancer, antimicrobial, anti-inflammatory, and antioxidant activities of extracts E. cuneata aerial parts, as well as to analysis of fatty acids ???? composition and phenolic acids by GC-Mass and HPLC.''
FATTY ACIDS HAVE NOT BEEN ANALYZED IN THE ARTICLE!
Please clarify this!
Author Response
"please see the attachment file"

Round 2
Reviewer 1 Report
I appreciated the efforts made by the authors to improve the manuscript. All required changes have been made and the manuscript is updated appropriately.
Reviewer 2 Report
I maintain that this paper has serious weaknesses and is currently insufficient and too preliminary for publication in Antioxidants.
I also still have serious doubts also about the HPLC method described here.